# Trends and Disparities in Self-Reported and Measured Osteoporosis among US Adults, 2007–2014

**DOI:** 10.3390/jcm8122052

**Published:** 2019-11-22

**Authors:** Qing Wu, Yingke Xu, Ge Lin

**Affiliations:** 1Nevada Institute of Personalized Medicine, University of Nevada, Las Vegas, NV 89154, USA; yingke.xu@unlv.edu; 2Department of Environmental and Occupational Health, School of Public Health, University of Nevada, Las Vegas, NV 89154, USA; ge.kan@unlv.edu

**Keywords:** self-reported, trend, health disparities, osteoporosis, NHANES

## Abstract

(1) Background: Studies examining osteoporosis trends among US adults by different socioeconomic status (SES) are limited. The prevalence of self-reported osteoporosis in the US is rarely reported. (2) Methods: Data from the National Health and Nutritional Examination Survey (NHANES) between 2007–2008 and 2013–2014 cycles were analyzed. Age-adjusted prevalence of self-reported and that of measured osteoporosis were calculated overall and by sex, race/ethnicity, education attainment, and SES. (3) Results: The prevalence of self-reported osteoporosis was higher than that of measured osteoporosis in all three survey cycles for women, and in 2007–2008 and 2009–2010 for men. Participants with high school/GED or higher educational attainment had an increased prevalence of measured osteoporosis during the study period. Among all SES groups, participants with low family income (PIR < 1.3) had the highest prevalence of measured osteoporosis, and the prevalence increased from 49.3 per 1000 population to 71.8 per 1000 population during the study period. (4) Conclusions: The prevalence of self-reported osteoporosis was higher than that of measured osteoporosis in US adults between 2007 and 2014. The age-adjusted prevalence of measured osteoporosis increased in participants with the educational attainment of high school/GED or above, and individuals with low family income.

## 1. Introduction

Osteoporosis is a significant public health problem because of a high number of associated fractures, as well as related morbidity, mortality, and decreased quality of life [1]. In the past 30 years, prospective cohort studies and national surveys have shown various trends of osteoporosis and fracture in US older adults [2,3,4,5]. A declining trend of hip fracture incidence was observed from Medicare data between 1985 and 2012, but the trend became plateaued in 2013–2014 [5]. A study that analyzed data from National Health and Nutrition Examination Surveys (NHANES) showed that the prevalence of osteoporosis at the femur neck declined between 1988–1994 and 2005–2006 [2]; however, a different study revealed the increased prevalence of osteoporosis at the same skeleton region from 2006 to 2014 [4]. The bone health status of Americans appears to have changed over the past decades. Discovering the trend of osteoporosis in recent years is vital for the prevention of this common, silent, costly disease and the associated devastating outcomes of fracture.

Most NHANES studies examining osteoporosis trends used the T-score method to define osteoporosis (T-score ≤ −2.5), but the trend of self-reported osteoporosis has not been studied yet. A prior study has demonstrated that the agreement between self-report and physical diagnosis was reasonable [6]. Self-reported estimates of diabetes, asthma, arthritis, and mental health conditions are increasingly used for policy and planning purposes [7]. “Self-reporting” was used to identify individuals with osteoporosis in a previous study as it is an economical and practical method [8]. Besides, the self-reported data reflect the perception of osteoporosis among the population. However, there is a paucity of studies comparing self-reported and measured prevalence of osteoporosis. Hence, the comparison between the self-reported and measured prevalence of osteoporosis can be informative. 

On the other hand, the relevant literature showed that health disparities exist across a range of clinical settings [9,10]. Studies examining disparities of osteoporosis in the US multiethnic population are limited, and the distribution and trend of self-reported osteoporosis in the US population remain unknown. Therefore, we aimed to examine the recent trends in self-reported and measured osteoporosis overall and by gender, race/ethnicity, and socioeconomic status (SES) in US adults, as well as a comparison of the prevalence of self-reported and measured osteoporosis during 2007 to 2014. 

## 2. Methods

### 2.1. Study Population

NHANES data from 2007–2008 through 2013–2014 were used in this study. NHANES has been conducted by the National Center for Health Statistics and Centers for Disease Control and Prevention (CDC) since 1999, by use of a complex, multistage probability sampling design. The plan of operation and sampling scheme are extensively described elsewhere [11]. Briefly, the NHANES sample population is the noninstitutionalized US civilian population of all ages residing in all 50 states and Washington DC. The survey examines a nationally representative sample of about 5000 people each year, all located in various counties across the country. To produce reliable statistics, NHANES oversamples persons 60 and older, African Americans, and Hispanics. Sample weights in NHANES have been constructed to adjust for non-response, oversampling, and non-coverage. Although portions of the household interviews and the health examination components have varied during its history, much of the survey and its anthropometry component methodology have remained consistent over time [12]. The use of consistent data collection methods in representative population samples makes this research invaluable in studying the trends and disparities in self-reported and measured osteoporosis across survey cycles. NHANES 2011–2012 does not have BMD measurements, and hence it was excluded from this study; thus, data of three survey cycles (2007–2008, 2009–2010, and 2013–2014) were analyzed in this study. In 2013–2014, NHANES only included participants aged 40 or older for the BMD measurement, while people aged eight or older were eligible for the BMD measurement during 2007–2010. We restricted the analysis to adults 40 years or older to ensure comparability across the three survey cycles. A separate sensitivity to examine the osteoporosis trends and disparities was conducted when including participants whose age 20 years and above. We also conducted an additional sensitivity analysis by including only postmenopausal women who are susceptible to osteoporosis. Women were categorized as postmenopausal if they answered “menopause” to “What is the reason that you have not had a period in the past 12 months?” This study was granted an exemption from the Institutional Board Review of the University of Nevada, Las Vegas, as we use de-identified, publicly available data for this research. (1004670–1)

### 2.2. Variables

The demographic characteristics, including age, gender, and racial/ethnicity, were ascertained by questionnaire. Bone mineral density (BMD) was measured using dual-energy X-ray absorptiometry (DXA). The femur scans in 2007–2008 through 2009–2010 were performed with a Hologic QDR-4500A fan-beam densitometer (Hologic, Inc., Marlborough, MA, USA), and the corresponding software was Hologic Discovery v12.4 (Hologic, Inc., Marlborough, MA, USA). In 2013–2014, the equipment and software used for measuring BMD had been updated, and the scans were conducted on Hologic Discovery model A densitometer (Hologic, Inc., Marlborough, MA, USA), with corresponding Apex 3.2 software (Hologic, Inc., Marlborough, MA, USA) [13]. Rigor quality control was maintained throughout the DXA data collection and scan analyses [13]. Additionally, the Hologic Service Team performed a cross-calibration procedure to standardize the updated system to the legacy system [14]. Moreover, a prior study assessed five femur regions and confirmed that there was no difference between mean BMD analyzed by Discovery 12.4 (Hologic, Inc., Marlborough, MA, USA) and by Apex 4.0 at the femur neck [4]. In NHANES, BMD measurement was performed at five regions of interest in the femur; however, only the femur neck was proposed by the World Health Organization (WHO) to be used as the skeletal site for the description of osteoporosis in populations [15]. Therefore, in the present study, BMD at the femur neck was used to calculate the T-score. Measured osteoporosis was defined as T-score ≤ −2.5. The NHANES had numerous questionnaires regarding bone health and osteoporosis. Self-reported osteoporosis was defined with an affirmative response to the questions, “Has a doctor ever told you that you had osteoporosis, sometimes called thin or brittle bones?” or “Were you ever treated for osteoporosis?” 

Educational attainment and family poverty income ratio (PIR) of participants were used as indicators of SES. Educational attainment was classified as less than high school, high school graduate/GED, some college, and college graduate or above [16]. The PIR is based on a comparison of family income with the poverty threshold determined by the US Bureau of Census. The PIR values were stratified into three categories: PIR < 130% (low income), 130% ≤ PIR ≤ 349% (middle income), and ≥350% (high income) [17].

### 2.3. Data analysis

Sampling weight was used to account for the complex survey design (e.g., unequal probabilities of selection) during analysis. The analysis was conducted among eligible participants who received a physical examination, and participants were excluded if they lacked valid BMD data. Age-adjusted prevalence (per 1000 population) of self-reported and measured osteoporosis were stratified by gender, race/ethnicity, educational attainment, and family income. Additionally, the 95% confidence intervals in the three survey cycles were calculated with linear regression. The age-adjusted prevalence estimates were derived by the direct method, using the 2000 US Census population as the standard population [18]. Standard errors, which were employed to construct confidence intervals, were estimated using Taylor series linearization. The analysis of variance (ANOVA) was used to test differences between groups for continuous variables, and a chi-square test was used to test the difference for categorical variables between groups. Orthogonal contrast was utilized to test the linear trend among the three survey cycles in the analysis. The difference between each self-reported value and the corresponding measured value was computed. Data analysis was conducted using SAS 9.4 (SAS Institute, Cary, NC, USA).

## 3. Results

### 3.1. Characteristics of Participants

In the current study, a total of 8151 eligible participants across NHANES 2007–2008 to 2013–2014 were included for the analysis, and their characteristics are presented in Table 1. From 2007–2008 to 2013–2014, the mean age of participants increased from 56.48 ± 0.32 years to 57.42 ± 0.30 years (*p* < 0.0001). Non-Hispanic white was the predominant population among the three racial/ethnic groups and made up more than 75% of participants in each survey cycle. During the three survey cycles, the BMI increased from 28.47 kg/m^2^ to 29.11 kg/m^2^ (*p* < 0.0001) while the BMD decreased from 0.803 g/cm^2^ to 0.782 g/cm^2^ (*p* < 0.0001). During the study period, the percentage of participants in each education level and in each PIR level were stable (all *p* > 0.2).

### 3.2. Disparities and Trends in the Prevalence of Self-Reported Osteoporosis

The overall age-adjusted prevalence of self-reported osteoporosis differed significantly by gender (*p* < 0.0001), and by race/ethnicity (*p* = 0.01) in NHANES from 2007–2008 to 2013–2014 (Table 2 and Figure 1A). During the three survey cycles, the age-adjusted prevalence of self-reported osteoporosis among men decreased from 20.3 per 1000 population in 2007–2008 to 13.7 per 1000 population in 2013–2014. In contrast, in women the age-adjusted prevalence decreased from 139.5 per 1000 population in 2007–2008 to 118.9 per 1000 population in 2009–2010, then increased to 142.5 per 1000 population in 2013–2014 (Figure 1A). In each racial/ethnicity group, women had a higher prevalence of self-reported osteoporosis than men during the three survey cycles (Figure 1B). In men, the age-adjusted prevalence of self-reported osteoporosis among Hispanic decreased from 25.8 per 1000 population in 2007–2008 to 18.7 per 1000 population in 2013–2014. Hispanic men had a higher prevalence of self-reported osteoporosis than other racial/ethnic men in each of the three survey cycles. In women, Non-Hispanic white had a higher prevalence of self-reported osteoporosis than other racial/ethnic groups: the prevalence increased from 147.1 per 1000 population to 155 per 1000 population during the study period. Across the three survey cycles, participants with college educational attainment had the greatest increase in age-adjusted prevalence of self-reported osteoporosis, with the prevalence increasing from 67.1 per 1000 population in 2007–2008 to 85.9 per 1000 population 2013–2014. However, those with less than high school attainment had the greatest decrease in age-adjusted prevalence of self-reported osteoporosis, with the prevalence decreasing from 96 per 1000 population in 2007–2008 to 67.3 per 1000 population in 2013–2014 (Figure 1C). Participants with low family income (PIR < 1.3) had the greatest decrease in the prevalence of self-reported osteoporosis, from 114.9 per 1000 population in 2007–2008 to 89.2 per 1000 population in 2013–2014 (Figure 1D). No significant linear trend was observed during the three survey cycles (all *p* for linear trend >0.05) in these subgroup analyses.

### 3.3. Disparities and Trends in Prevalence of Measured Osteoporosis

The age-adjusted prevalence of measured osteoporosis differed significantly by gender, race/ethnicity, and family income (Table 2, all *p* < 0.0001). The age-adjusted prevalence of measured osteoporosis increased from 47.5 per 1000 population in 2007–2008 to 63 per 1000 population in 2013–2014 among women. The prevalence also increased from 12.9 per 1000 population to 18.7 per 1000 population during the same period among men (Figure 1E). In the subgroup analysis of race/ethnicity, the age-adjusted prevalence in both genders of Non-Hispanic white had the greatest increase in the three survey cycles, with the prevalence increasing from 12.4 per 1000 population to 20 per 1000 population in men and from 48.4 per 1000 population to 70.4 per 1000 population in women (Figure 1F). Except for those with a less than high school diploma, the prevalence of measured osteoporosis in the other three groups with higher educational attainment increased through the three survey cycles (Figure 1G). For instance, the prevalence among participants with the highest educational attainment (college graduate or above) increased from 22.5 per 1000 population to 37 per 1000 population. Participants with low family income (PIR < 1.3) had the highest prevalence of measured osteoporosis over the three survey cycles, which increased steadily from 49.3 per 1000 population in 2007–2008 to 71.8 per 1000 population in 2013–2014 (Figure 1H, *p* = 0.06). No significant linear trend was observed in the prevalence of measured osteoporosis in all these subgroup analyses (all p for linear trend >0.05).

### 3.4. Prevalence of Self-Reported vs. Measured Osteoporosis

The age-adjusted prevalence of self-reported and measured osteoporosis in each of the three survey cycles for both men and women were compared in Table 3. In men, the prevalence of self-reported osteoporosis was higher than that of measured osteoporosis in 2007–2008 and 2009–2010, but not in 2013–2014. In women, the age-adjusted prevalence of self-reported osteoporosis was significantly higher than the corresponding prevalence of measured osteoporosis in all three survey cycles. There were 112 participants who had both self-reported osteoporosis and measured osteoporosis in the study. Among participants who had measured osteoporosis, 34% of them had self-reported osteoporosis.

We conducted two sensitivity analyses to further examine the disparities and trends in study samples with different inclusion criteria. The first sensitivity analysis included participants aged 20 or above, and the second sensitivity analysis included postmenopausal women only. The disparities and trends we observed were similar in the sensitivity analyses (Table A1, Table A2, Table A3 and Table A4).

## 4. Discussion

We used continuous NHANES, a nationally representative sample of the US civilian, noninstitutionalized population, to assess trends and disparities in measured and self-reported osteoporosis. Between 2007 and 2014, the age-adjusted prevalence of measured osteoporosis was on the rise among 40 year or older people in the United States, and this increase is not distributed uniformly across gender, race/ethnicity, and socioeconomic groups. Our findings regarding distinct trends and widening gaps by race/ethnicity and economic status are the concern, especially considering the significantly increased trend of measured osteoporosis among individuals with low family income (PIR < 1.3). Meanwhile, we found that the self-reported prevalence of osteoporosis was higher than the measured prevalence of osteoporosis. 

We found that the trend of age-adjusted prevalence of self-reporting is different and even opposite from that of measured osteoporosis during 2007–2014. Our finding was consistent with the study conducted by Dr. Stuart that the prevalence of self-reported osteoporosis tended to be higher than the prevalence of BMD defined osteoporosis [19]. Furthermore, the observed trend in prevalence of measured osteoporosis in this study also corresponded to the study by Looker et al., which reported that the prevalence of osteoporosis had an increasing trend in older (50 years and above) US adults between 2005–2006 and 2013–2014 [4]. These study findings are also consistent with our previous work in which we found a decreased trend of bone density in US adults 30 years and older between 2005 and 2014 [20]. The higher prevalence of self-reported osteoporosis might indicate the perception of osteoporosis in the population. Not only patients with osteoporosis, but also individuals with osteopenia or even healthy individuals, also had a perception of osteoporosis and bone health. Notably, the prevalence of self-reported osteoporosis was significantly higher than that of measured osteoporosis among women in all survey years, suggesting that more women may have a perception of this disease. Studies showed that the level of perception about the disease might modify individual behavior [21,22]. The Health Belief Model has confirmed that high perceptions provide an impetus to adopting health-protective behaviors [23]. Therefore, the perception of osteoporosis can positively impact people’s lifestyles and behavior. However, the perception could be one of the causes of a reduction in osteoporosis treatment as well. Related literature indicated that patients’ fear of adverse effects of bisphosphonates (medication for osteoporosis) might lead to the treatment crisis [24]. Thus, the inaccurate perception of osteoporosis might have a negative impact on their bone health, especially for patients with this condition. In addition, except for individuals with less than high school education, those with high school/GED or higher educational attainments had an increased prevalence of measured osteoporosis between 2007 and 2014. Individuals with higher educational attainment are more likely to engage in white-collar jobs, which require prolonged time spent sitting in the workplace [25]. Rapid advances in technology have led to increased numbers of employees with high education attainment being engaged in computerized tasks, which often require prolonged sedentary sitting; such sedentary behavior was associated with increased risk of osteoporosis. Studies found that full-time employees spent about half to two-thirds of their working hours sitting [26,27]. The increased number of white-collar employees affected by increased workplace sitting in the new technology era may explain the increased trend of osteoporosis in these groups with higher educational attainment. Workplace intervention to increase physical activity and to reduce sedentary time for white-collar workers may help to prevent osteoporosis in these groups. The prevalence of measured osteoporosis among participants with low family income (PIR < 1.3) increased significantly during the three survey cycles. Low income is always associated with disadvantaged SES [28], and has been considered as a major barrier for health care access. Individuals with low SES often have limited access to not only the healthcare system but also experience a shortage of healthy food and a lack of knowledge about health, ultimately leading to poor bone health. Rapid increases in health care costs and insurance premiums may also have contributed to widening disparities by income and SES.

Our analyses have limitations. First, the cross-sectional nature of NHANES limited our ability to assess the trajectory of BMD change over time for individual participants and the corresponding impacts on our findings. Second, some of the NHANES participants were not eligible for BMD testing due to hip fracture, pregnancy, or other reasons, and were excluded from analysis, which may cause bias in the estimates. Furthermore, non-response bias is always a concern in NHANES data, as response rates have declined in federal surveys since 2000 [29]. The decline in response rate could have a different impact on the accuracy of the estimated prevalence of self-reported osteoporosis across the different survey cycles we studied. However, the sample weights of NHANES have accounted for the non-response in the analysis. Third, the estimated prevalence of self-reported osteoporosis was subjected to recall bias. However, the NHANES survey was conducted in-person, which may be more accurate than a survey conducted by a different approach. Additionally, subjects were classified as self-reported osteoporosis in this study if they responded “yes” to either “Has a doctor ever told you that you had osteoporosis, sometimes called thin or brittle bones?” or “Were you ever treated for osteoporosis?” However, the difference between osteoporosis and osteopenia, and the detail osteoporosis treatment were not described in the NHANES questionnaires. Hence, the self-reported data were subject to bias. Finally, all of the NHANES participants were noninstitutionalized, as institutionalized persons may have lower bone mass [30], and the prevalence of measured osteoporosis in this study may be slightly underestimated. Nevertheless, this limitation is unlikely to have altered the trends of measured osteoporosis we observed. 

## 5. Conclusions

In summary, from 2007 to 2014, the prevalence of self-reported osteoporosis is higher than that of measured osteoporosis, and people with low income have a significant increasing trend in the prevalence of measured osteoporosis. Efforts to promote bone health in the population with low income are warranted in addressing such disparities in bone health. The age-adjusted prevalence of measured osteoporosis increased in participants with high school/GED or above education attainments. Effective interventions to promote healthy behaviors in these groups could be helpful. Additional research is warranted to further explain the different trends between self-reported and measured osteoporosis, as well as among different groups of race/ethnicity and SES in order to determine proper strategies to prevent osteoporosis and reduce disparities in bone health. The rapid growth of an increasingly diverse and aging US population marks a critical need to develop effective strategies to address these widening disparities in bone health.

## Figures and Tables

**Figure 1 jcm-08-02052-f001:**
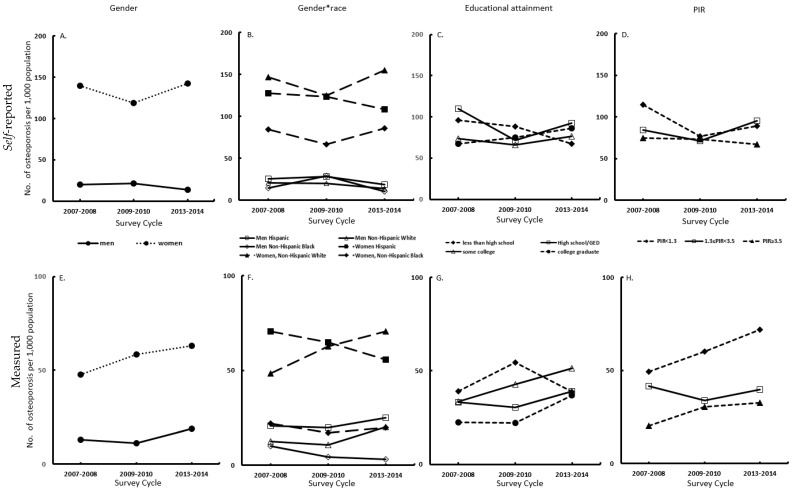
Trend in prevalence of self-reported and measured osteoporosis among participants ≥40 years or older in NHANES from 2007 to 2014. Abbreviations: PIR = Poverty Income Ratio.

**Table 1 jcm-08-02052-t001:** Characteristics of eligible participants aged 40 years or older in three National Health and Nutrition Examination Surveys (NHANES) from 2007 to 2014.

Characteristics	Survey Cycle			*p*-Value ^b^
	2007–2008 (*N* = 2790)	2009–2010 (*N* = 2903)	2013–2014 (*N* = 2458)	
Age, mean age (SD), years	56.48(0.32)	56.99 (0.40)	57.42 (0.30)	<0.0001
Gender, *N* (weighted%)				
Men	1412 (47.55)	1487 (49.20)	1222 (49.75)	0.2118
Women	1378 (52.45)	1416 (50.80)	1236 (50.25)	0.2118
Race, *N* (weighted%)				
Hispanic ^a^	712 (9.67)	804 (10.56)	593 (11.32)	0.8203
NH-white	1517 (80.25)	1601 (79.86)	1291 (77.91)	0.8262
NH-black	561 (10.08)	498 (9.58)	574 (10.77)	0.8215
Body mass index, mean (SD), kg/m^2^	28.47 (0.17)	28.62 (0.13)	29.11 (0.21)	<0.0001
Bone mineral density, mean (SD), g/cm^2^	0.803 (0.005)	0.800 (0.004)	0.782 (0.003)	<0.0001
Educational attainment, *N* (weighted%)				
<High school	860 (18.96)	864 (18.50)	566 (14.79)	0.2399
High school graduate/GED	680 (25.56)	668 (23.75)	569 (21.89)	0.2431
Some college	694 (27.90)	749 (27.73)	737 (30.96)	0.2882
≥college	556 (27.58)	622 (30.02)	586 (32.36)	0.4425
PIR, N (weighted%)				
<1.30	746 (15.66)	825 (16.20)	776 (19.16)	0.5254
1.30–3.49	1091 (33.27)	1117 (34.65)	858 (33.79)	0.8709
≥3.5	953 (51.07)	961 (49.15)	824 (47.05)	0.6403

Abbreviations: NH-white = Non-Hispanic white; NH-black = Non-Hispanic black. PIR = Poverty Income Ratio. ^a^ Hispanic includes Mexican American and other Hispanic. ^b^
*p*-value from the chi-square test comparing the difference between groups.

**Table 2 jcm-08-02052-t002:** Age-adjusted prevalence of self-reported and measured osteoporosis at the femur neck in adults aged 40+ years, across NHANES 2007–2014.

Characteristics	Self-Reported Prevalence, Per 1000 Population (95% CI)	*p*-Value ^b^	Measured Prevalence, Per 1000 Population (95% CI)	*p*-Value ^b^
Gender				
Men	21.6 (16.7, 26.5)	<0.0001	16.6 (13.1, 20.2)	<0.0001
Women	134.3(115.9, 152.7)		56.5 (48.1, 64.9)	
Race				
Hispanic ^a^	70.7 (59.0, 82.4)	0.01	37.23 (29.0, 45.5) ^d^	<0.0001
NH-white	83.5 (71.1, 95.9)		39.2 (33.5, 44.9)	
NH-black	55.6 (44.6, 66.5) ^c^		19.9 (14.5, 25.3) ^c^	
Educational attainment				
<High school	83.6 (69.7, 97.5)	0.10	45.4 (34.3, 56.5)	0.06
High school graduate/GED	91.6 (74.7, 108.6)		34.3 (26.1, 42.6)	
Some college	72.1 (56.0, 88.3)		43.7 (32.0, 55.5)	
≥college	74.1 (62.3, 86.0)		27.9 (19.7, 36.2)	
PIR				
<1.30	93.2 (72.7,113.6)	0.05	60.5 (49.3, 71.7)	<0.0001
1.30–3.49	85.3 (71.9, 98.7)		38.1 (29.7, 46.5) ^e^	
≥3.5	70.3 (60.0, 80.7)		28.1 (21.7, 34.5) ^f^	

Abbreviations: NH-white = Non-Hispanic white; NH-black = Non-Hispanic black. PIR = Poverty Income Ratio. ^a^ Hispanic includes Mexican American and other Hispanic. ^b^
*p*-value from chi-square test comparing the different characteristics between groups. ^c^
*p* < 0.01 for pair-wise comparison between NH-black and NH-white. ^d^
*p* < 0.01 for pair-wise comparison between NH-black and Hispanic. ^e^
*p* < 0.01 for pair-wise comparison between medium family income (1.3 ≤ PIR < 3.49) and low income (PIR < 1.3). ^f^
*p* < 0.01 for pair-wise comparison between high family income (PIR > 3.5) and low family income (PIR < 1.3).

**Table 3 jcm-08-02052-t003:** Prevalence comparison between self-reported and measured osteoporosis of the men and women in NHANES from 2007 to 2014.

Prevalence, Per 1000 Population	Survey Cycle		
	2007–2008 (*N* = 2790)	2009–2010 (*N* = 2903)	2013–2014 (*N* = 2458)
Men			
Self-reported, (95% CI)	20.3 (8.24, 32.4)	21.6 (14.6, 28.6)	13.7 (7.6, 19.8)
Measured Prevalence, (95% CI)	12.9 (6.5, 19.4)	10.9 (6.0, 15.8)	18.7 (12.6, 24.7)
Difference	7.4	10.7	−5.0
Women			
Self-reported, (95% CI)	139.5 (107.0, 171.9)	118.9 (99.1, 138.8)	142.4 (103.5, 181.5)
Measured, (95% CI)	47.5 (36.2, 58.7)	58.4 (44.2, 72.6)	63.0 (45.7, 80.3)
Difference	92.0	60.5	79.4

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
