# Peer review of "Trends and Disparities in Self-Reported and Measured Osteoporosis among US Adults, 2007–2014"

_jcm, 2019, doi:10.3390/jcm8122052_

Round 1

Reviewer 1 Report

The authors have done a great job studying the effect of various factor on self-reporting of osteoporosis. This is an important topic and would provide interesting basis for future studies.

In the methods section, it is mentioned that the self-reporting is extracted based on the questions on "Has a doctor ever told you that you had osteoporosis, sometimes called thin or brittle bones?" or "Were you ever treated for osteoporosis?" ... Considering the fact that many studies have shown the low awareness of some clinicians about osteoporosis and that some patients are diagnosed with and/or treated for osteoporosis without having a DXA, it would have been of additional value if the authors report how many of these self-reported cases corrolated with the DXA reports. It is also beneficial to define what do you mean by osteoporosis treatment, are the ca and vit d supplements also considered as treatment? And if there has been a question in NHANES questionnaire about "how this previous diagnosis was made, based on DXA or X-ray or ..", this would also be of great interest. 

I am also wondering would the results be different if the subjects would have been postmenopausal women rather than women over 40, or if the effect of the studied factors would have been studied in male and female separately.

Author Response

Comments to the Authors

The authors have done a great job studying the effect of various factors on self-reporting of osteoporosis. This is an important topic and would provide interesting basis for future studies.

Authors’ response: Thank you for the comments.

In the methods section, it is mentioned that the self-reporting is extracted based on the questions on "Has a doctor ever told you that you had osteoporosis, sometimes called thin or brittle bones?" or "Were you ever treated for osteoporosis?" ... Considering the fact that many studies have shown the low awareness of some clinicians about osteoporosis and that some patients are diagnosed with and/or treated for osteoporosis without having a DXA, it would have been of additional value if the authors report how many of these self-reported cases corrolated with the DXA reports.

Authors’ response: Thank you for your advice. In our study, 112 participants had both self-reported osteoporosis and measured osteoporosis, 34% of T-score defined osteoporosis patients reported with osteoporosis. We added the corresponding content to the results section in the revision. (Please see Prevalence of self-reported vs. measured osteoporosis under Results).

It is also beneficial to define what do you mean by osteoporosis treatment, are the ca and vit d supplements also considered as treatment?

Authors’ response: Thank you for the suggestion. According to the documentation of NHANES, the participants were asked, "Ever treated for osteoporosis?” during the survey. However, the NHAENS questionnaires were not designed to collect detailed information about the osteoporosis treatment, including what type of medication the participants used. Therefore, no information was available regarding the definition of osteoporosis treatment of NHANES, which is a common limitation for the secondary data analysis research; we have acknowledged this limitation in the discussion section of the manuscript. (Please see the third paragraph of Discussion). 

And if there has been a question in NHANES questionnaire about "how this previous diagnosis was made, based on DXA or X-ray or ..", this would also be of great interest.

Authors’ response: Thank you for the question, and we agree with the reviewer this questions is of great interest. To address the reviewer's concern, we reviewed all related questionnaires and documentation of NHANES and found that there is no such question available in NHANES questionnaires.

I am also wondering would the results be different if the subjects would have been postmenopausal women rather than women over 40 or if the effect of the studied factors would have been studied in male and female separately.

Authors’ response: Thank you for your suggestions. In the revision, we followed the reviewers' recommendations and conducted a sensitivity analysis among postmenopausal women, and the results were similar to the main analysis for women over 40. In this sensitivity analysis, the overall age-adjusted prevalence of self-reported osteoporosis differed significantly by race/ethnicity (p= 0.01) in NHANES from 2007-2008 to 2013-2014, and the age-adjusted prevalence of measured osteoporosis differed significantly by both race/ethnicity (p<0.0001), and family income (p<0.05). Also, the age-adjusted prevalence of self-reported osteoporosis was significantly higher than the corresponding prevalence of measured osteoporosis in all three survey cycles. We added these new analyses in the revised manuscript. (Please see study population under Methods, Prevalence of self-reported vs. measured osteoporosis under Results and supplementary table A3 and Table A4).

Reviewer 2 Report

Excellent article. Solid data, well presented and sound conclusions 

Of great interest to your readership

Author Response

Comments to the Authors

Excellent article. Solid data, well presented and sound conclusions of great interest to your readership

Authors’ response: Thanks for your comments.

Reviewer 3 Report

Review of Wu, Xu and Lin, “Trends and Disparities I’m Self-Reported and Measured Osteoporosis among US Adults, 2007-2014”

This MS examines the relationship between self-reported and measured osteoporosis in US adults using the NHANES, with the goal in determining how accurate self-reporting is and if there are race, gender, and SES disparities. Overall, the authors found that self-reporting was higher than measured osteoporosis, and this was to a greater extent in women. Low family income also resulted in the highest prevalence of measured osteoporosis, as did attainment of high school/GED education or above. Overall, the study was reasonably designed and executed. However, its unclear how useful this information will be. Perhaps the authors should also describe situations where self-reported osteoporosis has been used. A few additional issues need to be addressed:

In Table 2 as well as in the appendix, it is not clear whether the p-values relate to self-reported vs. measured or characteristics, such as male vs female. Also, if they do relate to characteristics, it would be useful to show which pair-wise comparisons were different. In Table A2, it looks like there might be typos in the CI’s. For example, 7.1 (2.6, 1.16) and 23.6 (1.5, 32.1). When were the questionnaires regarding bone health administered? Was it before or after DXA measurement? Was there any concern that enrollment in the study might influence the self-reporting? For example, if someone had previously gotten a DXA scan based on suspicion of low bone density/osteoporosis, would they be more likely to self-report osteoporosis? Also, it may be useful to determine if patients with osteopenia often mis-reported, stating that they had osteoporosis? Were patients informed as to the exact definitions of osteopenia and osteoporosis? 

Author Response

Comments to the Authors

This MS examines the relationship between self-reported and measured osteoporosis in US adults using the NHANES, with the goal in determining how accurate self-reporting is and if there are race, gender, and SES disparities. Overall, the authors found that self-reporting was higher than measured osteoporosis, and this was to a greater extent in women. Low family income also resulted in the highest prevalence of measured osteoporosis, as did attainment of high school/GED education or above. Overall, the study was reasonably designed and executed.

Authors’ response: Thank you.

However, its unclear how useful this information will be. Perhaps the authors should also describe situations where self-reported osteoporosis has been used.

Authors’ response:

 Thanks for your suggestion. In a previous study (Australian Institute of Health and Welfare: A snapshot of osteoporosis in Australia 2011), self-report data were used to identify patients with osteoporosis because it is an economical and efficient way. Also, the self-reported data could reflect people's perception of this disease; the higher prevalence of self-reported osteoporosis might indicate a better perception of osteoporosis in the population. In the present study, the prevalence of self-reported osteoporosis was significantly higher than that of measured osteoporosis among women in all survey years, suggesting that more women may have a perception of this condition. Following the reviewer’s suggestions, we have modified the corresponding contents in the revised manuscript. (Please see the second paragraph under Introduction and the second paragraph under Discussion).

In Table 2 as well as in the appendix, it is not clear whether the p-values relate to self-reported vs. measured or characteristics, such as male vs female. Also, if they do relate to characteristics, it would be useful to show which pair-wise comparisons were different.

Authors’ response: Thank you for the comments. The p-values were from chi-square tests for comparing the different characteristics between groups. In the revision, we have added detailed information and the results of pair-wise comparisons to clarify this in Table 2.

In Table A2, it looks like there might be typos in the CI’s. For example, 7.1 (2.6, 1.16) and 23.6 (1.5, 32.1).

Authors’ response: Thank you for pointing out the typos. We corrected the typo “(2.6, 1.16)” to “(2.6, 11.6)” in Table A2. However, there is no typo in “23.6 (1.5, 32.1).”

When were the questionnaires regarding bone health administered? Was it before or after DXA measurement? Was there any concern that enrollment in the study might influence the self-reporting? For example, if someone had previously gotten a DXA scan based on suspicion of low bone density/osteoporosis, would they be more likely to self-report osteoporosis?

Authors’ response:

Thanks for your comments. NHANES collects the data of the questionnaire as well as the health examination. Usually, participants need to sign consent forms for participating in the interview and physical examination. The questionnaire is distributed during the home interview; the physical examination (including DXA measurement) is conducted at the Mobile Exam Center (MEC) within 1 to 2 weeks after the interview. To address the reviewer's concern, we searched the relevant documents in NHANES, and no information indicated that the participant had DXA measurement before this survey.

Also, it may be useful to determine if patients with osteopenia often mis-reported, stating that they had osteoporosis? Were patients informed as to the exact definitions of osteopenia and osteoporosis? 

Authors’ response:

 We agree with the reviewer that recall bias could happened in the survey study. We have acknowledged this limitation “the estimated prevalence of self-reported osteoporosis was subjected to recall bias” in the discussion section of this study. To further address the reviewer’s concern, we reviewed all related documents of NHANES. However, no information is available if all patients were provided the exact definition of osteopenia or osteoporosis, which is one of the limitations of secondary data analysis; we acknowledged this limitation in the revision. (Please see the third paragraph of Discussion).